# Investigation of Feature Engineering Methods for Domain-Knowledge-Assisted Bearing Fault Diagnosis

**DOI:** 10.3390/e25091278

**Published:** 2023-08-30

**Authors:** Christoph Bienefeld, Florian Michael Becker-Dombrowsky, Etnik Shatri, Eckhard Kirchner

**Affiliations:** 1Institute for Product Development and Machine Elements, Technical University of Darmstadt, Otto-Berndt-Straße 2, 64287 Darmstadt, Germany; florian_michael.becker@tu-darmstadt.de (F.M.B.-D.); kirchner@pmd.tu-darmstadt.de (E.K.); 2Bosch Center for AI, Corporate Research, Robert Bosch GmbH, Robert-Bosch-Campus 1, 71272 Renningen, Germany

**Keywords:** bearing fault diagnosis, feature engineering, machine learning, condition monitoring, frequency band separation

## Abstract

The engineering challenge of rolling bearing condition monitoring has led to a large number of method developments over the past few years. Most commonly, vibration measurement data are used for fault diagnosis using machine learning algorithms. In current research, purely data-driven deep learning methods are becoming increasingly popular, aiming for accurate predictions of bearing faults without requiring bearing-specific domain knowledge. Opposing this trend in popularity, the present paper takes a more traditional approach, incorporating domain knowledge by evaluating a variety of feature engineering methods in combination with a random forest classifier. For a comprehensive feature engineering study, a total of 42 mathematical feature formulas are combined with the preprocessing methods of envelope analysis, empirical mode decomposition, wavelet transforms, and frequency band separations. While each single processing method and feature formula is known from the literature, the presented paper contributes to the body of knowledge by investigating novel series connections of processing methods and feature formulas. Using the CWRU bearing fault data for performance evaluation, feature calculation based on the processing method of frequency band separation leads to particularly high prediction accuracies, while at the same time being very efficient in terms of low computational effort. Additionally, in comparison with deep learning approaches, the proposed feature engineering method provides excellent accuracies and enables explainability.

## 1. Introduction

In rotating machinery, it is becoming increasingly important to avoid unforeseen rolling bearing failures. This trend is driven by both cost efficiency and sustainability. To detect bearing damage at an early stage, sophisticated condition monitoring is required. For this purpose, a selection must be made from a variety of possible measurands [1]. According to the current state of research, vibration measurements are the most commonly used signals for condition monitoring of rotating machinery. Therefore, vibrations are usually measured with acceleration sensors under operating conditions. The information contained in the acceleration signals can then be extracted to monitor the condition of the bearing, which provides the foundation for a predictive maintenance strategy [2].

To infer condition information from raw acceleration signals, a variety of data processing techniques are available in the literature. Machine learning methods in particular are being used more and more frequently to evaluate large amounts of data in a targeted manner. A distinction must be made between deep learning and more traditional machine learning (ML) methods, with the latter often used in combination with upstream data processing—so-called feature engineering. More recently, methodological research on fault diagnosis using vibration signals has moved very much towards purely data-driven deep learning approaches [3]. Motivation for these deep learning approaches is that only little specific expert or domain knowledge is required by the practitioner for data evaluation. Furthermore, for big data applications, it is expected that deep learning models can provide better predictive accuracies than more traditional approaches [4]. However, the disadvantage of purely data-driven approaches is that the path of decision making, starting with the data and ending with the fault diagnosis, is hardly explainable.

In contrast to purely data-driven deep learning approaches, more classical approaches combine feature engineering methods with machine learning models [5]. By incorporating domain knowledge and computing deterministic features, explainability can be obtained. Furthermore, due to the use of preprocessed feature data, one can rely on computationally efficient machine learning algorithms, which reduces the training effort compared with most deep learning approaches.

The remainder of this paper is structured in the following way: Section 2 presents fundamentals of the CWRU dataset and a collection of feature engineering methods. Fundamentals of popular processing methods and a list of relevant feature formulas are provided. Subsequently, specific feature extraction methods are presented in Section 3, deriving novel feature calculation techniques by systematically applying series connections of processing methods and feature formulas. Section 4 discusses the resulting fault diagnosis accuracies obtained using the different feature sets. Furthermore, the results are analyzed with respect to deep-learning-based results from the literature. To analyze feature explainability, visualizations of feature characteristics are presented. The final Section 5 critically summarizes the research findings.

## 2. Fundamentals

Literature provides public bearing datasets for the investigation of different fault diagnosis methods. A well-structured literature overview of fault diagnosis methods and the datasets used can be found in publications by Schwendemann et al. [6] and Hakim et al. [7].

### 2.1. CWRU Bearing Fault Data

An extensive dataset is provided by Case Western Research University (CWRU) with a variety of ball bearing damages and operating conditions [8]. Due to its wide scope, it is particularly well suited for the evaluation of fault diagnosis algorithms and is used very frequently in the literature [9]. For this reason, it will also be applied for evaluation of the methods presented in this paper. For the CWRU data, different sizes of damage are introduced to the inner ring, outer ring and rolling element on different bearings. In addition, undamaged bearings are also included for reference. With these prepared bearings, vibration signals are measured on a test bench under different speeds and engine loads. For further details, the description and analysis of the CWRU dataset by Smith and Randall [10] is recommended.

### 2.2. Feature Formulas

To extract features from the raw vibration signals, mathematical feature formulas can be applied to the acceleration signal. A distinction is made between the feature calculation in the time domain and in frequency domain. Lei et al. [11] propose a set of 11 time-domain and 13 frequency-domain features, adding an additional 14th frequency domain feature in [12]. Based on the recommendations by Lei et al. and further literature on this topic, the features gathered in Table 1 and Table 2 were identified for further use throughout this article. The time-domain features are calculated using the discrete amplitude values of the vibration signal *x* and its length *N*.

For calculating the features in the frequency domain, the signal is transformed typically via fast Fourier transform (FFT). To avoid spectral leakage, windowing of the time signal has to be performed before applying FFT [19]. For the investigations carried out here, a Hanning window will be implemented for this purpose. For the mathematical formulations of the frequency-domain features, the vector of discrete amplitudes in the frequency domain *s* is used. Furthermore, the frequencies *f* associated with the amplitudes and the number of discrete frequencies *M* are required for calculation.

### 2.3. Signal Processing Methods

The features presented in the previous section can be applied directly to the raw signal and its Fourier transform, as described before. However, it is also common to use additional processing methods and perform a feature calculation based on such preprocessed signals [12].

#### 2.3.1. Envelope Analysis

With the help of envelope analysis, relevant information can be extracted from the measurement signal, which is particularly important for the detection of initial rolling bearing failures [2]. Rolling bearing damage leads to periodic shock excitations. However, these are short, decay very quickly and often times are superimposed with non-roller-bearing vibration excitations, making them difficult or impossible to detect using frequency analysis of the raw signal. The high-frequency carrier signal is amplitude-modulated by the frequency of the low-frequency modulation signal of the vibration excitations [20]. For the diagnosis of rolling bearing damage, it is usually not the carrier signal that is of interest but the modulation signal, which can be extracted by demodulation. Envelope analysis can be implemented using the Hilbert transform, causing demodulation and consequently allowing identification of shocks and analysis of characteristic damage [21]. In envelope analysis, the envelope is first determined from the raw signal. In a further step, the envelope spectrum can be determined via Fourier transform.

#### 2.3.2. Empirical Mode Decomposition

The empirical mode decomposition (EMD) is based on the assumption that signals can be decomposed into different intrinsic mode functions (IMFs). The iterative process for determining the IMFs is called sifting. Spline interpolation of the signals local maxima and minima is an essential part of the sifting process. For further details, Nandi and Ahmed [2] provide a comprehensive description of the EMD procedure. The sifting process can be stopped by a termination criterion. One simple termination criterion can be given by predefining the number of IMFs to be decomposed. After decomposition, the original signal can be rearranged as the sum of all decomposed IMFs and its residual.

#### 2.3.3. Wavelet Transform

The wavelet transform (WT) is a well-known signal processing strategy for analyzing nonstationary and transient signals in the time–frequency domain [22]. Using the wavelet transform, data can be decomposed into different frequency components, which are then analyzed with a resolution adapted to their scale [23]. In contrast to the Fourier transform, the wavelet transform uses wavelet basis functions that contain several frequencies [24]. A wavelet family consisting of different wavelet basis functions is formed by scaling and translation of a mother wavelet. Within wavelet transforms, continuous wavelet transform (CWT), discrete wavelet transforms (DWT) and wavelet packet transform (WPT) can be distinguished [2]. Since in the context of this work discrete measurement data are used and the computational effort is kept low, the focus here is on DWT and WPT.

For DWT, first, the measurement signal is put through complementary low-pass and high-pass filters [25]. The low-frequency signal is called approximation coefficient, and the high-frequency signal is called detail coefficient. In the next iteration step, the approximation coefficient is again decomposed into a high-frequency and low-frequency component with simultaneously decreasing resolution. The number of iteration steps can be adjusted by setting the decomposition level. In contrast to DWT, where only the low-frequency signal is split at each further decomposition level, WPT also splits the high-frequency signal into new high-frequency and low-frequency components [2].

#### 2.3.4. Frequency Bands

A particularly computationally efficient processing method has already been proposed in [26] and further investigated in [27]. The core of this method is to separate the signal into frequency bands. For this purpose, the amplitude spectrum of the signal is first determined via FFT. Subsequently, this spectrum is split into bands. Figure 1 shows frequency band separation using eight equally sized frequency bands as an example.

Originally, the use of equally sized frequency bands and mean values calculated from them was proposed in [26]. Within the present paper, the frequency band method will additionally be extended to octave- and third-octave-based frequency band sizes. Furthermore, besides calculating the mean value only, all the feature formulas in the frequency domain shown in Table 2 will be applied.

## 3. Methods

The primary goal of this paper is to identify the best-performing feature engineering methods by comparison. For this reason, all other factors potentially influencing the results within this comparison must remain unchanged. These constant factors include the dataset and the machine learning specification, as well as the evaluation of the results using chosen metrics.

### 3.1. Data Preparation

In addition to comparing the different feature engineering approaches to each other, it is of additional interest to determine how well the proposed methods perform in comparison with deep learning approaches from the literature. For this reason, the data processing used in the present work must strictly follow an approach used in the literature. A well-documented data processing pipeline and classification task is used by Magar et al. [28], who propose a Convolutional Neural Network (CNN) approach for fault diagnosis. In the present paper, both the data processing and the classification task are adopted therefrom. Hence, the Seeded Fault Test Data provided by Case Western Research University [8] are used. Only the 48 kHz Drive End Bearing Fault Data with 2 hp motor load are processed. The tested bearings are SKF deep-groove ball bearings of type 6205-2RS JEM. In total, data from 9 different bearing damages and 1 undamaged bearing are investigated. Based on the original data naming scheme by CWRU [8], the bearings are labeled as follows:Inner ring faults: IR007_2, IR014_2 and IR021_2;Outer ring faults: OR007@6_2, OR014@6_2 and OR021@6_2;Ball faults: B007_2, B014_2 and B021_2;No fault: Normal_2.

Using these data, the machine learning task is to clearly distinguish between the 10 different classes. Following the example of Magar et al. [28], 467,600 values are used for each bearing data, each of which is divided into packages of 1670 values. This results in 280 instances for each class, for which the feature calculation can be performed.

### 3.2. Machine Learning

A random forest is used as the machine learning algorithm for the studies presented here. Random forests belong to the tree-based machine learning methods. According to Grinsztajn et al. [29], they are superior to neural networks for small-to-medium-sized tabular data. Also, Fernandez-Delgado et al. [30] showed the superiority of random forests in a comparison of different classification algorithms based on a large number of datasets. For the present case, the random forest is parametrized with a maximum tree depth of 20 and a tree count of 500. Training and testing of the random forest is performed within a stratified 5-fold cross-validation. This allows for increased statistical significance and consideration of the uncertainties in the results obtained. Since the different bearing fault classes are equally distributed within the data, and all classes can be considered of equal value weighting, the accuracy of the prediction on the test data is chosen as the metric. For overall performance evaluation, the mean value of the accuracies within cross-validation is used.

### 3.3. Feature Engineering

Using the presented data and machine learning procedure, different feature sets will be generated and reduced, and the achieved prediction accuracies will be compared. The systematic comparison of different feature engineering methods is based on the modular design visualized in Figure 2.

The modules can be combined in various ways. A complete analysis of all possible method combinations is not realizable due to effort limitations. Therefore, the primary task of the investigations will be to compare the performance of each single signal processing methods with each other. Thus, one single signal processing method is always considered per feature set. The following step-by-step approach is used to build the feature sets examined in the remainder of this paper:Selection of an appropriate set of feature formulas based on the raw, unprocessed vibration signal (RAW).Comparison of the different processing methods using the feature formulas selected in the previous step.Additional investigations of the frequency bands: Consideration of the frequency-domain mean values solely, as proposed in [26].

Based on this approach, a total of 19 feature sets are formed. These feature sets, including their detailed settings, are listed in Table 3. The settings of the processing methods were determined during preliminary investigations.

It is apparent that the total number of features varies between 10 and 336 for the different feature sets. To ensure that the comparisons are not biased by the feature count, the following 3 approaches are applied to all feature sets:All features of the calculated feature set are used for the evaluation of the prediction accuracy—*Complete feature set*.Based on the random forest feature importance evaluated on the complete features set, the 10 most important features are selected and used to evaluate prediction performance—*10 most important features (RFFI)*.The feature sets are transformed using a principal component analysis (PCA), and only the 10 principal components representing the largest feature variance are used to evaluate prediction performance—*10 principal components (PCA)*.

The workflow thus realized for evaluating the different feature sets is presented in Figure 3.

## 4. Results and Discussion

In the following, the prediction accuracies achieved using the different feature engineering methods are compared. The results are shown in Figure 4, where each bar indicates the mean value of the accuracies obtained within the five-fold cross-validation. Additionally, the standard deviation resulting from the individual cross-validation iterations is visualized at the top of each bar. For each feature set, the complete feature set is used, as well as the feature sets reduced to 10 features with the help of the RFFI and PCA methods for comparison purposes.

Looking at the results generated using the raw signal directly, it is apparent that neither the sole use of the time-domain (RAW_TD) nor the frequency-domain (RAW_FD) features leads to particularly good results. In comparison, combining the two domains leads to significantly better performance. However, the complete collection of 42 features presented in the fundamentals section (RAW_all) cannot provide a significant advantage compared to the 25 features proposed by Lei et al. [11,12] (RAW_Lei). Therefore, in order to generate compact feature sets, the feature set according to Lei et al. is used for further investigations. Applying envelope analysis (ENV) does not show any advantage over the raw signal in the analyses performed here. However, both Empirical Mode Decompostion (EMD) and Wavelet Transforms (DWT and WPT) lead to increased accuracies when considering the complete feature set. What is remarkable about the latter feature sets is their rather bad performance when using the reduced 10 principal components. Apparently, due to the large feature count of these feature sets, relevant information is lost when applying PCA.

Especially good results are provided by all feature sets based on the frequency band processing methods (FB and OFB). A differentiation is to be made between feature sets that use all 14 frequency domain features and the feature sets that are formed exclusively with the frequency-domain mean value. Both variants work very well, and the accuracies reach values close to 100%. The following four feature sets achieve the highest average accuracies with values above 99.9%:FB_20_FD;FB_100_FD-mean;OFB_one-octave_FD;OFB_third-octave_FD.

Furthermore, some of the feature sets reduced to a feature count of 10 achieve average accuracies above 99.0%. The two best-performing feature sets reduced using the RFFI method are listed first:FB_20_FD-mean;OFB_third-octave_FD-mean.

Reducing the number of features to 10 by PCA, the following two feature sets perform best:FB_20_FD;OFB_third-octave_FD.

Based on these results, it can be concluded that considering separated frequency bands for feature calculation is particularly good for extracting the entirety of relevant information from vibration signals.

To further compare the results with a deep learning method, the performance evaluation procedure was based on Magar et al. [28]. They propose a deep CNN for the classification of rolling bearing faults and achieved an average accuracy of 98.5% with their best-performing three channel CNN. This result is significantly exceeded by the methods presented here—especially when frequency band separation is applied as the signal processing method.

To evaluate the computed feature sets also in terms of their computational effort, the computation times are measured and visualized in Figure 5.

The first thing to notice here is that EMD is a very computationally expensive method, which can be explained by the high number of computational steps required when using the iterative algorithm. Feature sets that are based on frequency bands (FB) as processing method and the mean frequency-domain value (FD-mean) as feature formula require particularly low computational effort. Since they provide very good prediction results at the same time, the use of this feature engineering method seems to be recommendable.

The actual training of the random forest requires similarly little computation time, like the feature generation. A single training process of the random forest takes between 0.8 and 1.3 s. The slight differences in training duration can be explained by the different number of features per set. In terms of the required computational effort, the feature-engineering-based approach proposed in this paper clearly offers advantages over purely data-driven deep learning algorithms that are trained based on the raw signals.

Since the approach using frequency band separations is recommended, the features based on frequency bands are now discussed in more detail. For visualization purposes, the feature set FB_20_FD-mean is used here due to its compactness. To show the importance of each feature based on this feature set, the random forest feature importance is shown in Figure 6. Both the mean values represented by the bars, as well as the standard deviations resulting from cross-validation, can be seen.

From this diagram, the importance of the frequency components for condition monitoring can be observed. There are very important frequency bands both in the low-frequency range around 2 kHz and in the high-frequency range towards 24 kHz. According to feature importance, some frequency bands in the middle frequency ranges contain only a small amount of information.

The bearing characteristic excitation frequencies for the considered CWRU experiments are within the range of 11 Hz to 156 Hz and are thus found in the lowest frequency band of FB_20. These characteristic frequency calculations are performed according to the equations given by Randall et al. [21]. We assume that the particularly important frequency bands contain exactly the frequencies whose associated test rig eigenmodes are particularly strongly excited in the case of certain bearing faults. Thereby, different damages excite different modes due to their different characteristic frequencies. For vibration-based condition monitoring, the detection of exactly these eigenfrequencies is of particular importance.

To illustrate what distinguishes an important from an unimportant feature, the value distributions of the most important and least important features from the feature set FB_20_FD-mean are shown as box plots over the different bearing faults in Figure 7.

A well-performing and important feature is generally characterized by the fact that the value distributions can be used to separate the different classes as well as possible. In this example, an almost complete separation can be made purely visually between the classes B007_2, Baseline_2, IR007_2 and OR007_6_2 based on the feature values for the important feature (left). Based on the unimportant feature (right), the distributions of the feature values overlap much more. Therefore, a clear separation of the different classes is only possible to a minor extent. Such a domain-knowledge-supported feature analysis allows understanding the decision making of the machine learning algorithm. In purely data-driven approaches, such analysis is hardly possible, which makes the case for the use of explicit feature engineering from an explainability perspective.

## 5. Conclusions

This article demonstrated that the fault diagnosis of rolling bearings can be optimized by applying proficient feature engineering incorporating domain knowledge. For this purpose, known feature generation methods from various literature sources and novel method combinations were investigated. Applied to the example of CWRU bearing fault data, feature sets generated based on frequency band separation performed particularly well, achieving cross-validated accuracies above 99.9%. Given the presented results, we recommend using the FB_20_FD feature set without feature reduction as an exemplary benchmark for future investigations. Besides the extension of investigations using different datasets, it would be of interest for future research to evaluate an even larger feature engineering design space.

In addition to the excellent prediction accuracy, the proposed feature engineering approach provides advantages due to the simplicity of the calculation and the associated low computational cost. Furthermore, compared with purely data-driven approaches, the feature engineering method offers a good starting point for the interpretation of features and thus improved explainability. Compared to another publication which examined the same dataset using a deep CNN method, a significant improvement in prediction accuracy was achieved. It should be critically noted here that the CWRU dataset provides a rather small amount of data. We suspect that purely data-driven deep learning methods may be more effective on much larger datasets.

To conclude, the most complex deep learning approaches should not necessarily be used for vibration-based classification of rolling bearing faults. As demonstrated, feature engineering methods based on frequency band separation and rather simple machine learning algorithms like random forests can enable very high prediction accuracies, especially on small-to-medium-sized data. In particular, we propose to consider the features based on frequency band separation for benchmarking purposes when developing or applying more complex deep learning methods.

## Figures and Tables

**Figure 1 entropy-25-01278-f001:**
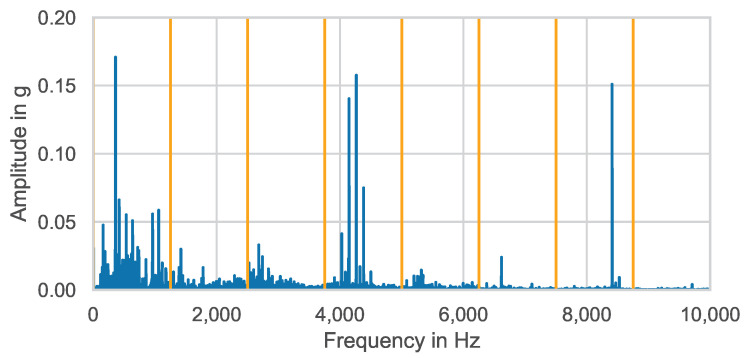
Equally sized frequency band separation (orange) of amplitude spectrum (blue).

**Figure 2 entropy-25-01278-f002:**
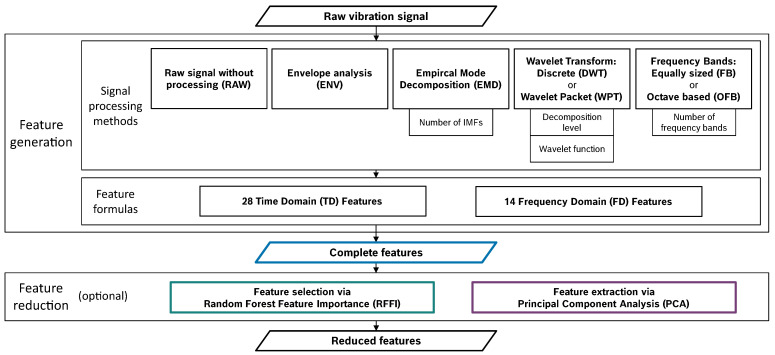
Modular design of feature engineering.

**Figure 3 entropy-25-01278-f003:**
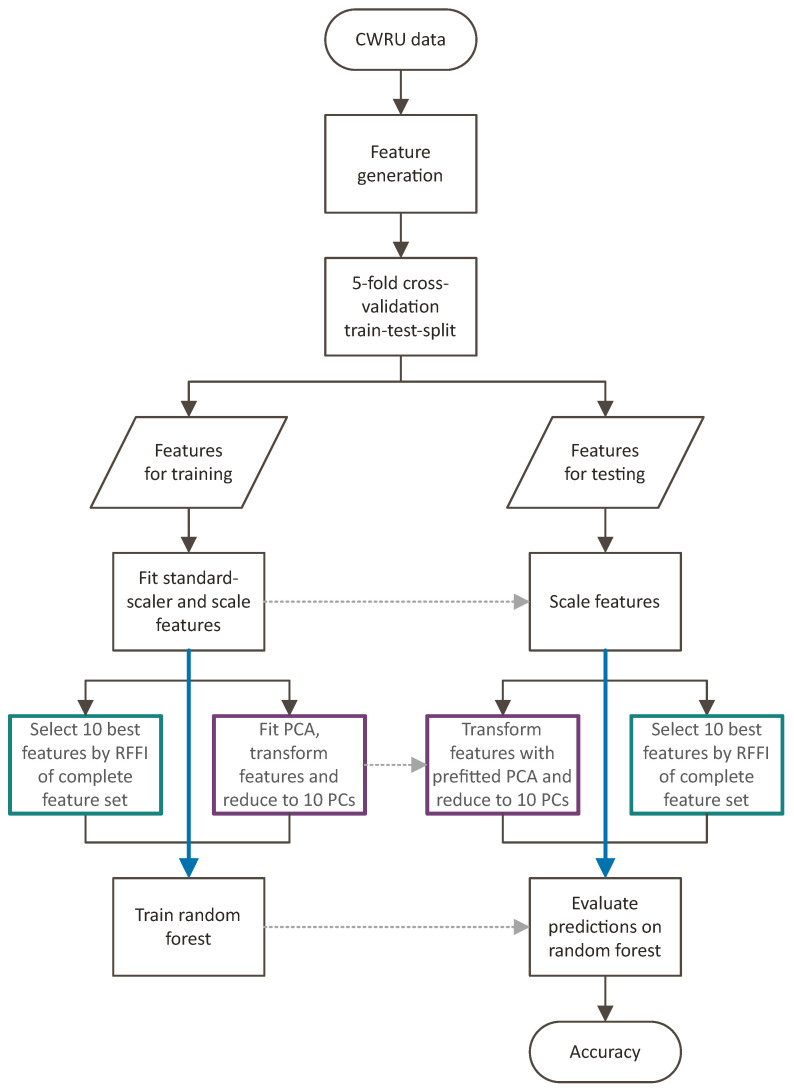
Workflow for feature set evaluation.

**Figure 4 entropy-25-01278-f004:**
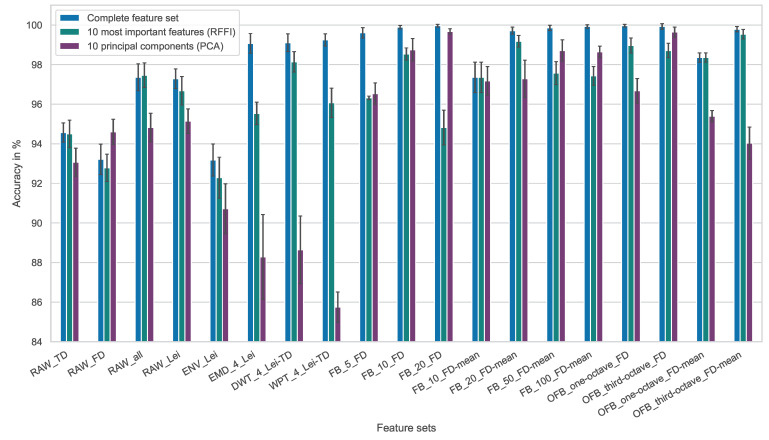
Accuracy comparison for different features sets.

**Figure 5 entropy-25-01278-f005:**
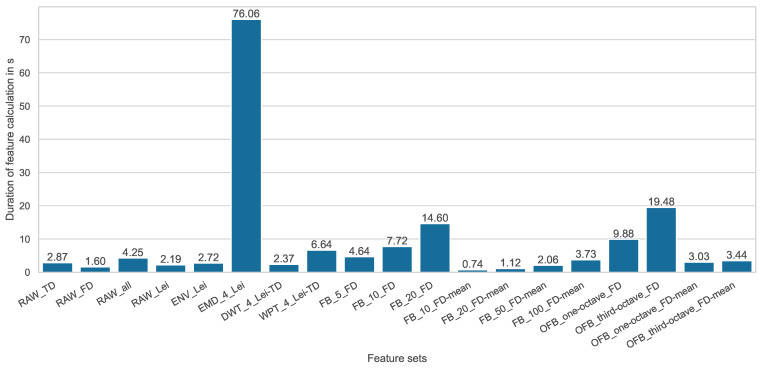
Feature calculation timings for different features sets.

**Figure 6 entropy-25-01278-f006:**
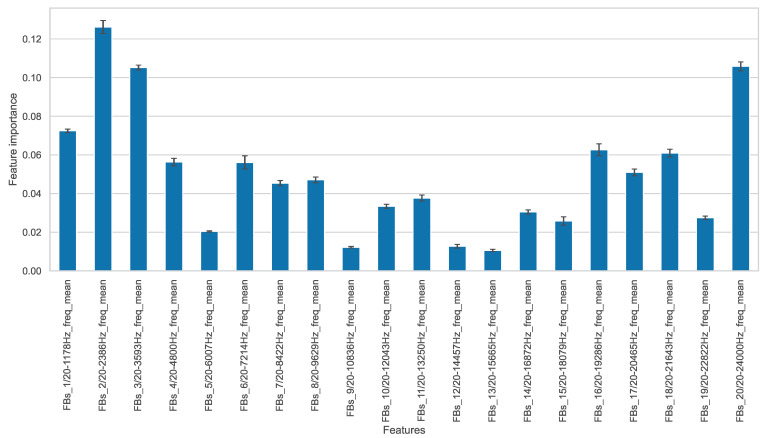
Random forest feature importance for feature set FB_20_FD-mean.

**Figure 7 entropy-25-01278-f007:**
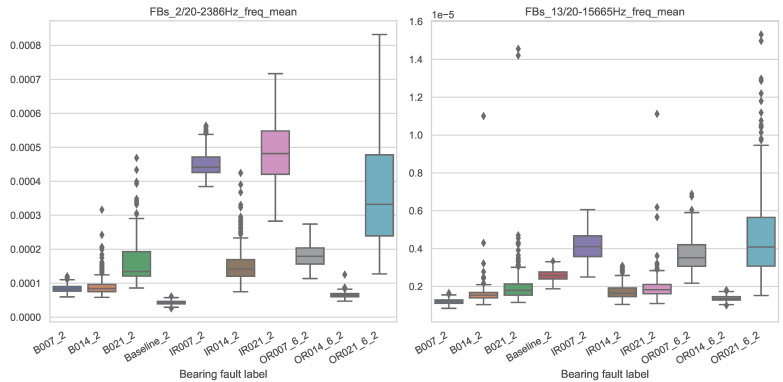
Box plot of feature value distributions over the different classes for the most important feature (**left**) and least important feature (**right**) from feature set FB_20_FD-mean.

**Table 1 entropy-25-01278-t001:** Feature formulas in time domain.

Feature	Formula	
Mean	$T_{1}=\frac{1}{N}\sum_{i=1}^{N}x_{i}$	[11]
Standard deviation	$T_{2}=\sqrt{\frac{1}{N-1}\sum_{i=1}^{N}{\left(x_{i}-T_{1}\right)}^{2}}$	[11]
Square root mean (SRM)	$T_{3}={\left(\frac{1}{N}\sum_{i=1}^{N}\sqrt{|x_{i}|}\right)}^{2}$	[11]
Root mean square (RMS)	$T_{4}=\sqrt{\frac{1}{N}\sum_{i=1}^{N}x_{i}^{2}}$	[11]
Maximum absolute	$T_{5}=\max\left(|x|\right)$	[11]
Skewness	$T_{6}=\frac{\sum_{i=1}^{N}{\left(x_{i}-T_{1}\right)}^{3}}{\left(N-1\right)\cdot T_{2}^{3}}$	[11]
Kurtosis	$T_{7}=\frac{\sum_{i=1}^{N}{\left(x_{i}-T_{1}\right)}^{4}}{\left(N-1\right)\cdot T_{2}^{4}}$	[11]
Crest factor	$T_{8}=\frac{T_{5}}{T_{4}}$	[11]
Clearance indicator	$T_{9}=\frac{T_{5}}{T_{3}}$	[11]
Shape indicator	$T_{10}=\frac{T_{4}}{\frac{1}{N}\sum_{i=1}^{N}\left|x_{i}\right|}$	[11]
Impulse indicator	$T_{11}=\frac{T_{5}}{\frac{1}{N}\sum_{i=1}^{N}\left|x_{i}\right|}$	[11]
Skewness factor	$T_{12}=\frac{T_{6}}{T_{4}^{3}}$	[13]
Kurtosis factor	$T_{13}=\frac{T_{7}}{T_{4}^{4}}$	[13]
Mean absolute	$T_{14}=\frac{1}{N}\sum_{i=1}^{N}\left|x_{i}\right|$	[14]
Variance	$T_{15}=\frac{1}{N}\sum_{i=1}^{N}{\left(x_{i}-T_{1}\right)}^{2}$	[14]
Peak	$T_{16}=\frac{\max(x)-\min(x)}{2}$	[14]
K factor	$T_{17}=T_{16}\cdot T_{4}$	[14]
Energy	$T_{18}=\sum_{i=1}^{N}x_{i}^{2}$	[15]
Mean absolute deviation	$T_{19}=\frac{1}{N}\sum_{i=1}^{N}\left|x_{i}-T_{1}\right|$	[16]
Median	$T_{20}=\mathrm{median}\left(x_{i}\right)$	[16]
Median absolute deviation	$T_{21}=\mathrm{median}(|x_{i}-T_{20}\left|\right)$	[16]
Rate of zero crossings	$T_{22}=\frac{\mathrm{number}\mathrm{of}\mathrm{zero}\mathrm{crossings}}{N}$	[16]
Product RMS kurtosis	$T_{23}=T_{4}\cdot T_{7}$	[17]
Fifth moment	$T_{24}=\frac{\sum_{i=1}^{N}{\left(x_{i}-T_{1}\right)}^{5}}{T_{2}^{5}}$	[18]
Sixth moment	$T_{25}=\frac{\sum_{i=1}^{N}{\left(x_{i}-T_{1}\right)}^{6}}{T_{2}^{6}}$	[18]
RMS shape factor	$T_{26}=\frac{T_{4}}{T_{14}}$	[18]
SRM shape factor	$T_{27}=\frac{T_{3}}{T_{14}}$	[18]
Latitude factor	$T_{28}=\frac{\max(x)}{T_{3}}$	[18]

**Table 2 entropy-25-01278-t002:** Feature formulas in frequency domain.

Feature	Formula	
Mean	$F_{1}=\frac{\sum_{j=1}^{M}s_{j}}{M}$	[11]
Variance	$F_{2}=\frac{\sum_{j=1}^{M}{\left(s_{j}-F_{1}\right)}^{2}}{M-1}$	[11]
Third moment	$F_{3}=\frac{\sum_{j=1}^{M}{\left(s_{j}-F_{1}\right)}^{3}}{M\cdot {\left(\sqrt{F_{2}}\right)}^{3}}$	[11]
Fourth moment	$F_{4}=\frac{\sum_{j=1}^{M}{\left(s_{j}-F_{1}\right)}^{4}}{M\cdot F_{2}^{2}}$	[11]
Grand mean	$F_{5}=\frac{\sum_{j=1}^{M}f_{j}\cdot s_{j}}{\sum_{j=1}^{M}s_{j}}$	[11]
Standard deviation 1	$F_{6}=\sqrt{\frac{\sum_{j=1}^{M}{\left(f_{j}-F_{5}\right)}^{2}\cdot s_{j}}{M}}$	[11]
C Factor	$F_{7}=\sqrt{\frac{\sum_{j=1}^{M}f_{j}^{2}\cdot s_{j}}{\sum_{j=1}^{M}s_{j}}}$	[11]
D Factor	$F_{8}=\sqrt{\frac{\sum_{j=1}^{M}f_{j}^{4}\cdot s_{j}}{\sum_{j=1}^{M}f_{j}^{2}\cdot s_{j}}}$	[11]
E Factor	$F_{9}=\frac{\sum_{j=1}^{M}f_{j}^{2}\cdot s_{j}}{\sqrt{\sum_{j=1}^{M}s_{j}\sum_{j=1}^{M}f_{j}^{4}\cdot s_{j}}}$	[11]
G Factor	$F_{10}=\frac{F_{6}}{F_{5}}$	[11]
Third moment 1	$F_{11}=\frac{\sum_{j=1}^{M}{\left(f_{j}-F_{5}\right)}^{3}\cdot s_{j}}{M\cdot F_{6}^{3}}$	[11]
Fourth moment 1	$F_{12}=\frac{\sum_{j=1}^{M}{\left(f_{j}-F_{5}\right)}^{4}\cdot s_{j}}{M\cdot F_{6}^{4}}$	[11]
H Factor	$F_{13}=\frac{\sum_{j=1}^{M}\sqrt{f_{j}-F_{5}}\cdot s_{j}}{M\cdot \sqrt{F_{6}}}$	[11]
J Factor	$F_{14}=\sqrt{\frac{\sum_{j=1}^{M}{\left(f_{j}-F_{5}\right)}^{2}\cdot s_{j}}{\sum_{j=1}^{M}s_{j}}}$	[12]

**Table 3 entropy-25-01278-t003:** Feature set explanations.

Feature Set Name	Processing Method	Settings of the Processing Method	Feature Formulas	Complete Feature Count
RAW_TD	Raw signal	-	28 time-domain features: $T_{1}$ to $T_{28}$	28
RAW_FD	Raw signal	-	14 frequency-domain features: $F_{1}$ to $F_{14}$	14
RAW_all	Raw signal	-	All 42 features: $T_{1}$ to $T_{28}$ and $F_{1}$ to $F_{14}$	42
RAW_Lei	Raw signal	-	25 features according to Lei et al.: $T_{1}$ to $T_{11}$ and $F_{1}$ to $F_{14}$	25
ENV_Lei	Envelope analysis	-	25 features according to Lei et al.: $T_{1}$ to $T_{11}$ and $F_{1}$ to $F_{14}$	25
EMD_4_Lei	Empirical mode decomposition	Number of extracted IMFs: 4	25 features according to Lei et al.: $T_{1}$ to $T_{11}$ and $F_{1}$ to $F_{14}$	125
DWT_4_Lei-TD	Discrete wavelet transform	Decomposition level: 4 Wavelet: Daubechies 13	11 time-domain features according to Lei et al.: $T_{1}$ to $T_{11}$	55
WPT_4_Lei-TD	Wavelet Packet Transform	Decomposition level: 4 Wavelet: Daubechies 13	11 time-domain features according to Lei et al.: $T_{1}$ to $T_{11}$	176
FB_5_FD	Equally sized frequency bands	Number of frequency bands: 5	14 frequency -domain features: $F_{1}$ to $F_{14}$	70
FB_10_FD	Equally sized frequency bands	Number of frequency bands: 10	14 frequency -domain features: $F_{1}$ to $F_{14}$	140
FB_20_FD	Equally sized frequency bands	Number of frequency bands: 20	14 frequency domain features: $F_{1}$ to $F_{14}$	280
FB_10_FD-mean	Equally sized frequency bands	Number of frequency bands: 10	1 feature: Mean value in frequency domain: $F_{1}$	10
FB_20_FD-mean	Equally sized frequency bands	Number of frequency bands: 20	1 feature: Mean value in frequency domain: $F_{1}$	20
FB_50_FD-mean	Equally sized frequency bands	Number of frequency bands: 50	1 feature: Mean value in frequency domain: $F_{1}$	50
FB_100_FD-mean	Equally sized frequency bands	Number of frequency bands: 100	1 feature: Mean value in frequency domain: $F_{1}$	100
OFB_one-octave_FD	Octave based frequency bands	Frequency band size: One octave	14 frequency-domain features: $F_{1}$ to $F_{14}$	140
OFB_third-octave_FD	Octave based frequency bands	Frequency band size: Third octave	14 frequency-domain features: $F_{1}$ to $F_{14}$	336
OFB_one-octave_FD-mean	Octave-based frequency bands	Frequency band size: One octave	1 feature: Mean value in frequency domain: $F_{1}$	10
OFB_third-octave_FD-mean	Octave-based frequency bands	Frequency band size: Third octave	1 feature: Mean value in frequency domain: $F_{1}$	24

## Data Availability

Data were obtained from Case Western Reserve University and are available at https://engineering.case.edu/bearingdatacenter (accessed on 7 March 2023).

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
