# Peer review of "Investigation of Feature Engineering Methods for Domain-Knowledge-Assisted Bearing Fault Diagnosis"

_entropy, 2023, doi:10.3390/e25091278_

Round 1
Reviewer 1 Report
The paper presents results of an engineered feature selection in combination with random forest for the deagnosis of earing faults.
The paper is well written.
Choices appear a little bit unexplained completely and why some choices have been taken is not always well stated. I.e., why random forest and not any other ML approach. Is it the best performing? Why the authors chose a certain number of bands in signal decomposition? What filters and why the wavelet transform use in this application? Why in the scoring of results standard parameters like Precision, Recall, accuracy, etc are not used in this case and instead some other measures are produced?
The discussion of the obtained results appears to me somehow limited and should be stressed a bit more to allow a more fair comparison with other papers addressing the same problem.
Limited review is required for the English, in my opinion.
Author Response
Response to Reviewer 1 Comments
Point 1: Choices appear a little bit unexplained completely and why some choices have been taken is not always well stated. I.e., why random forest and not any other ML approach. Is it the best performing?
Response 1: As written in section 3, for feature investigation reasons all other factors potentially influencing the results must remain unchanged. The machine learning algorithm itself is one of those potential influencing factors. Why we chose a random forest in particular is stated in section 3.2. Given the state of research, a random forest provides a reliable and very well performing algorithm for the given kind of tabular data. Additional investigations of machine learning algorithms and their hyperparameters would make the investigation space explode in size. Thus, based on literature and our own preliminary investigations, we made the choice for random forest as single machine learning algorithm.
Point 2: Why the authors chose a certain number of bands in signal decomposition? What filters and why the wavelet transform use in this application?
Response 2: The different sizes of frequency bands where chosen to investigate a wide variety of settings while still keeping the total amount of features within a reasonable count. In combination with the selected feature formulas the total feature count can kind of explode for some setting combinations. Although some choices simply had to be made, we always had the goal to enable a fair comparison by generating feature sets that are representative for the best possible performance of each method combination.
Point 3: Why in the scoring of results standard parameters like Precision, Recall, accuracy, etc are not used in this case and instead some other measures are produced?
Response 3: Additional clarifications have been added in chapter 3.2 regarding the choice of accuracy as metric. Additionally, I want to explain it here: The evaluation metric was chosen to be the accuracy only. Reason for this choice is that the classes inside the selected dataset are equally distributed (using 467600 values per bearing data as stated in line 171) and neither false negativ nor false positive predictions have to be weighted in some particular fashion. Thus, the use of Precision and Recall does not provide any significant benefit.
Point 4: The discussion of the obtained results appears to me somehow limited and should be stressed a bit more to allow a more fair comparison with other papers addressing the same problem.
Response 4: We have added some further content for discussion and conclusion, especially providing some more details for future research recommendations.

Reviewer 2 Report
Considering the large number of features and the number of comparisons made, the authors could present a recommended procedure for performing a similar analysis as the one presented in the article. A flowchart similar to Fig. 2 would be desirable, but with the conclusions drawn. This will make it easier for the reader to proceed when analyzing data for another case. Specifically, what steps should be followed and what to pay special attention to in a given step.
Author Response
Response to Reviewer 2 Comments
Point 1: Considering the large number of features and the number of comparisons made, the authors could present a recommended procedure for performing a similar analysis as the one presented in the article. A flowchart similar to Fig. 2 would be desirable, but with the conclusions drawn. This will make it easier for the reader to proceed when analyzing data for another case. Specifically, what steps should be followed and what to pay special attention to in a given step.
Response 1: The conclusion part has been edited and expanded to give a more specific recommendation for future research. Hopefully, this clarifies the recommended procedure without the need of adding a separate figure reusing contents from Figure 2.

Round 2
Reviewer 2 Report
The answers to my questions are satisfactory. The article may be published in its current form.